# Complexing Protein-Free Botulinum Neurotoxin A Formulations: Implications of Excipients for Immunogenicity

**DOI:** 10.3390/toxins16020101

**Published:** 2024-02-10

**Authors:** Michael Uwe Martin, Juergen Frevert, Clifton Ming Tay

**Affiliations:** 1Independent Researcher, 31832 Springe, Germany; 2Merz Therapeutics GmbH, 60318 Frankfurt, Germany; 3Merz Asia Pacific Pte., Ltd., Singapore 138567, Singapore; clifton.tay@merz.sg

**Keywords:** botulinum neurotoxin A, Coretox^®^, daxibotulinumtoxinA, excipients, immunogenicity, immunoresistance, incobotulinumtoxinA, secondary treatment failure

## Abstract

The formation of neutralizing antibodies is a growing concern in the use of botulinum neurotoxin A (BoNT/A) as it may result in secondary treatment failure. Differences in the immunogenicity of BoNT/A formulations have been attributed to the presence of pharmacologically unnecessary bacterial components. Reportedly, the rate of antibody-mediated secondary non-response is lowest in complexing protein-free (CF) IncobotulinumtoxinA (INCO). Here, the published data and literature on the composition and properties of the three commercially available CF-BoNT/A formulations, namely, INCO, Coretox^®^ (CORE), and DaxibotulinumtoxinA (DAXI), are reviewed to elucidate the implications for their potential immunogenicity. While all three BoNT/A formulations are free of complexing proteins and contain the core BoNT/A molecule as the active pharmaceutical ingredient, they differ in their production protocols and excipients, which may affect their immunogenicity. INCO contains only two immunologically inconspicuous excipients, namely, human serum albumin and sucrose, and has demonstrated low immunogenicity in daily practice and clinical studies for more than ten years. DAXI contains four excipients, namely, L-histidine, trehalosedihydrate, polysorbate 20, and the highly charged RTP004 peptide, of which the latter two may increase the immunogenicity of BoNT/A by introducing neo-epitopes. In early clinical studies with DAXI, antibodies against BoNT/A and RTP004 were found at low frequencies; however, the follow-up period was critically short, with a maximum of three injections. CORE contains four excipients: L-methionine, sucrose, NaCl, and polysorbate 20. Presently, no data are available on the immunogenicity of CORE in human beings. It remains to be seen whether all three CF BoNT/A formulations demonstrate the same low immunogenicity in patients over a long period of time.

## 1. Introduction

### 1.1. Natural Botulinum Neurotoxin A (BoNT/A) Comes as a Large Complex

The food poison BoNT/A is produced by the bacterium *Clostridium botulinum* (reviewed in [1]) as a large complex of different proteins of approximately 900 kDa, also called holotoxin or large progenitor toxin complex [2]. The proteins have different functions in causing food poisoning (botulism) by inhibiting neurotransmission at peripheral nerve terminals, resulting in flaccid paralysis.

The core neurotoxin BoNT/A in its bioactive form consists of two subunits—one heavy chain (100 kDa) and one light chain (50 kDa)—covalently linked by a disulfide bridge (Figure 1a). The heavy chain is the neuroselective domain that binds specifically to receptors at pre-synaptic nerve terminals and facilitates the uptake of the core BoNT/A into the recycling vesicle. From the acidified vesicle, the translocation domain of the heavy chain transports the light chain into the nerve terminal cytosol. The light chain is a metalloprotease that then cleaves SNAP-25, required for vesicle fusion and, thus, neurotransmitter release. This core molecule of 150 kDa is necessary and sufficient to inhibit the release of neurotransmitters at nerve terminals if it is injected into the target tissue. The other bacterial molecules do not participate in neuromodulation but play different roles if the holotoxin is ingested orally with contaminated food and subsequently crosses the intestinal barrier to enter the lymph and bloodstream. The non-toxic non-hemagglutinin protein (NTNHA, 150 kDa) links the 150 kDa BoNT/A non-covalently to the 12-subunit hemagglutinin complex [3,4]. Together, the NTNHA and hemagglutinins form a large progenitor complex of 900 kDa (schematically depicted in Figure 1b, based on [3]).

They protect the 150 kDa BoNT/A from digestion in the gastrointestinal tract. In addition, the **hemagglutinins** also facilitate the uptake of the core neurotoxin in the gut by interacting with the adhesion molecules present in the tight junctions of the gut epithelial barrier [4]. This allows the translocation of BoNT/A from the gut lumen into the blood and lymph. These additional proteins are also called complexing proteins (CPs) or non-toxic neurotoxin-associated proteins (NAPs). It is noteworthy that NTNHA is a pH sensor and that the binding of NTNHA to BoNT/A is dependent on pH [5]. In the acidic environment of the stomach and the still slightly acid environment of the upper gut lumen, the 900 kDa large progenitor complex is stable, and all proteins stay together. However, once the complex is delivered into the tissue of the upper gut, the pH changes from slightly acidic to neutral or even weakly basic (~pH, 7.4). This alteration in pH induces a conformational change in NTNHA that results in the release of the 150 kDa BoNT/A into the tissue fluid. This means that the large progenitor complex dissociates in a pH-dependent manner [6], and the 150 kDa BoNT/A makes its way within the body in the lymph or bloodstream alone, i.e., independent of the complexing proteins. This explains why the complexing proteins are not required in a pharmaceutical BoNT/A formulation as the pharma protein is injected into tissues rather than taken up through the gut epithelial barrier. As the dissociation of the large progenitor complex is dependent on the pH, this also occurs already in the vial as soon as the lyophilized or freeze-dried pharmaceutical is reconstituted in a neutral liquid, such as saline, during clinical application [7] or, latest, after the reconstituted material is injected into the tissue, where a pH of 7.4 Is found under normal conditions.

### 1.2. The Pharmaceutical Formulations of BoNT/A Differ in Purity and Excipients

Presently, all pharmaceutical companies produce BoNT/A by growing *Clostridium botulinum* type A in a fermenter using proprietary production protocols. As BoNT/A is purified from these bacterial cell cultures, all available BoNT/A preparations are natural bacterial proteins or “biologicals”. Although the principal processes of production and purification of the pharma protein by different manufacturers are comparable, they are not the same, and products differ in two important aspects. Firstly, in purity (see Figure 2), and secondly, in the excipients added to the neurotoxin (see below in Table 1).

All formulations contain the core neurotoxin, 150 kDa BoNT/A, and if the companies utilize the same bacterial strain, such as the Hall A strain of *Clostridium botulinum*, it can be assumed that the resulting 150 kDa BoNT/A molecules are similar or even identical. It should be noted that this is only an assumption as the amino acid sequences of BoNT/A in the different products have not been established (e.g., by mass spectrometry and sequencing) and compared to each other. The only published DNA sequence available is that for OnabotulinumtoxinA [19]. If bacterial strains other than the *Clostridium botulinum* Hall A strain are used for production, it cannot be excluded that the neurotoxin molecules differ in their amino acid sequences. Although the resulting proteins may still work as neurotoxins, this may have implications for the immunogenicity of the individual pharma protein. Most BoNT/A products available contain bacterial proteins in addition to the 150 kDa BoNT/A. These include complexing proteins in almost all products, as well as other bacterial molecules reported to be present in some products, e.g., flagellin or flagellin breakdown products in AbobotulinumtoxinA (ABO; Dysport^®^/Azzalure^®^, Ipsen Ltd., Slough, Berkshire, UK) [13,20] and bacterial DNA in OnabotulinumtoxinA (ONA; Botox^®^/Vistabel^®^, Allergan Inc., Irvine, CA, USA) (Figure 2) [21]. While there may have been historic reasons for retaining complexing proteins in a BoNT/A formulation, all additional bacterial components should be considered as contaminants (reviewed in [18]).

## 2. Immunological Consequences of Impurities in a BoNT/A Formulation

### 2.1. Importance of Purity for BoNT/A

BoNT/A in its pure and bio-active form, as contained in IncobotulinumtoxinA (INCO; Xeomin^®^/Bocouture^®^, Merz Pharmaceuticals GmbH, Frankfurt, Germany) [8,22], is of low immunogenicity and does not induce the formation of neutralizing antibodies (nAbs), resulting in antibody-mediated secondary non-response (SNR) if patients were treated exclusively with INCO (reviewed in [23]). This has been demonstrated by a series of well-controlled clinical studies [24,25,26]. However, the presence of complexing proteins and other bacterial contaminants have been discussed to affect the immunogenicity of BoNT/A formulations due to their adjuvant properties. This is true for therapeutic applications [27] and in aesthetic use [28]. It has been shown that repeated injections of complexing-protein-containing (CPC) BoNT/A formulations may lead to the formation of nAbs to BoNT/A [29]. This may result in partial or complete antibody-mediated SNR or treatment failure, also addressed as immunoresistance to BoNT/A. Immunoresistance was reported for at least two BoNT/A formulations (ONA and ABO), both containing bacterial components not required for the neuromodulatory effect of the pharmaceutical. In consequence, the purity of a BoNT/A formulation plays a role in immunogenicity, and the absence of complexing proteins and other bacterial contaminants reduces the risk of inducing nAbs and SNR to this valuable pharma protein.

### 2.2. Neutralizing Antibodies to BoNT/A

It is well accepted that the repeated injection of tetanus toxoid is a vaccination, with the production of protecting nAbs as a desired outcome. On the other hand, the induction of nAbs after the repeated injection of BoNT/A, very closely related to tetanus toxin, is not an intended effect and should be avoided as far as possible. In the field of clinical applications, e.g., in neurology, the risk of inducing nAbs that may result in SNR is documented by many clinical studies and more than 30 years of experience. Yet, the relevance of nAbs and SNR in aesthetic practice is still disputed, seemingly neglectable due to the low frequencies reported (reviewed in [30,31,32]). However, this perception has undergone a change in recent years, especially in Asia [33,34], where aesthetic patients are becoming younger, treatments more frequent, and the number of applications steadily increasing. In addition, doses of BoNT/A that are injected in one session, e.g., for body contouring, can now reach levels that were previously only known in therapeutic indications [35]. It needs to be stressed that once nAbs have developed in a patient, these will curtail future treatment options in therapeutic as well as aesthetic indications for many years [36]. Therefore, an awareness of the immunogenicity of different BoNT/A formulations is of importance. And, understanding why and how the immune system generates antibodies to some BoNT/A products but not to others presents an opportunity to reduce the risk of formation of nAbs. Recent studies have shown that there is a correlation between the presence of bacterial complexing proteins and possibly other bacterial contaminants with an increased risk of the formation of nAbs [29].

### 2.3. Why and How the Immune System Responds to Injections with BoNT/A

Antibodies are valuable tools of the immune system that can protect human beings from dangerous challenges, most prominently microbes such as viruses and bacteria. Yet, the production of antibodies is a time- and energy-consuming strenuous effort for the immune system, requiring the sequential activation of at least three different types of immune cells: dendritic cells (DCs), T helper lymphocytes (Th cells), and B lymphocytes (B cells). This tight control of events is necessary, first, to avoid the full activation of the immune system to harmless agents, such as nutrients in our food, and reserve it for truly dangerous challenges. Second, tight control prevents the generation of autoreactive antibodies that can cause autoimmune diseases. Dangerous challenges include pathogens like bacteria that can infect us, proliferate rapidly, compete for nutrients with our own cells, and intoxicate us with their products. Basically, two important decisions have to be made by two different types of immune cells. This happens in a strictly hierarchical order before the process of antibody production can commence. The first rapid decision in response to a challenge is made by DCs. These sentinel cells are distributed all over our body and detect “danger signals” (reviewed in [37,38]) with specialized pathogen pattern recognition receptors (reviewed in [39]). Most prominent microbial “danger signals” include prototypical surface molecules of bacteria, such as flagellin, lipopolysaccharides of Gram-negative bacteria, or peptidoglycans of Gram-positive bacteria, but also bacterial DNA. Once these receptors bind to such “danger signals”, DCs become activated and, together with tissue macrophages, initiate an acute local inflammation to combat the microbial challenge. In consequence, they phagocytose what they have recognized as being “**dangerous**” and digest it. This has two consequences: First, the phagocytosed microbe is killed and, thus, is unable to further multiply. Second, the digestion of microbial proteins yields peptides. These peptides are subsequently loaded onto specialized peptide-presenting molecules, also known as major histocompatibility complex proteins (MHCs) or human leukocyte antigens (HLAs). In parallel, the activated DCs move to the next draining lymph node where they show these peptides in the MHC molecules to the second decision maker, the T helper lymphocyte, in a process referred to as professional antigen presentation. Th cells specifically identify presented peptides and distinguish “**foreign** or non-self” from “own or self”, “foreign” being the second decision that has to be made. If a Th cell specifically recognizes the presented peptide, it becomes activated and subsequently supports antigen-specific B cells to become activated and finally produce antigen-specific antibodies. The schematic in Figure 3 simply summarizes how antibodies are made, and the strict hierarchy in decision making: first “dangerous” and then “foreign”, as antigen presentation is an absolute prerequisite for Th cell activation, and antigen presentation is only possible if the DC is fully activated by “danger signals”.

In consequence, Th-cell-dependent classical production of antibodies is impossible if there are no danger signals present, even if a foreign protein is ingested, e.g., when we eat. This is of great importance when trying to explain the differences in immunogenicity that have been observed when comparing BoNT/A products that contain bacterial components in addition to the 150 kDa BoNT/A neurotoxin with highly purified products that are free of contaminating bacterial components. The pure 150 kDA BoNT/A molecule is, of course, a bacterial protein. Hence, it is “foreign”, and peptides derived from BoNT/A can be readily detected by human T cells [40,41], and subsequently, B cells can produce antibodies specific for BoNT/A, even protective antibodies, as is well known from vaccination in the past (reviewed in [42,43]). However, BoNT/A is a protein that is produced within and released by the bacteria; it is not a surface structure such as flagellin. Thus, it is not a prototypic danger signal and, by itself, is unable to activate DCs. To date, only one report claims that 150 kDa BoNT/A is capable of activating mouse macrophages via TLR2 [44]. However, the purity of the BoNT/A preparation used in this study remains unclear. This is a major issue and concern, as it cannot be excluded that minute amounts of peptidoglycans, present in the walls of *Clostridium botulinum type A* [45], may have been present, which are excellent activators of TLR2 ([46] reviewed in [47]). Further, the specific potency of the preparation was very low (only 5% of commercial products), and the concentration of BoNT/A in the assays was very high (>1500 LD50 Units/mL). One can conclude that the findings have no clinical relevance.

Therefore, due to the lack of “danger signals”, DC will not be activated if pure 150 kDa BoNT/A in its bioactive form is introduced into the tissue. Pure and bioactive BoNT/A by itself is a poor or weak immunogen. However, if ligands for pathogen pattern recognition receptors are contained in a BoNT/A formulation, these can serve as “danger signals”, providing the pivotal activation signal for the DCs to become activated. Bacterial components that have been identified in some CPC-BoNT/A products and that can activate DCs include flagellin in ABO [13,20], clostridial DNA in ONA [21], hemagglutinins in all CPC BoNT/A products [48,49,50,51,52,53], and inactive neurotoxin [17]. These substances injected alone or in combination with 150 kDa BoNT/A at the same time and in the same place can serve as adjuvants, substances that are well known from vaccinology to increase the immunogenicity of “weak” immunogens (reviewed in [54]) by providing the first danger signal required to kickstart the immune cascade.

## 3. Complexing Protein-Free (CPF) BoNT/A Preparations

### 3.1. Three BoNT/A Formulations Are Free of Complexing Proteins but They Are Not the Same

Scientific research and clinical experience for more than ten years have proven that complexing proteins are not required for the neuromodulatory activity of injected BoNT/A. This and the established lower immunogenicity of CPF-INCO have prompted other manufacturers to further develop and produce highly purified BoNT/A formulations that are free of complexing proteins.

Presently (end of 2023), three BoNT/A formulations are available that are reported to be free of complexing proteins (see Table 1). These are (1) IncobotulinumtoxinA (INCO; Xeomin^®^/Bocouture^®^, Merz Pharmaceuticals GmbH, Frankfurt, Germany); (2) MT10107 (CORE; Coretox^®^, Medytox Inc., Cheongju-si, Seoul, South Korea); and (3) DaxibotulinumtoxinA-lanm (DAXI; DAXXIFY^TM^, Revance Therapeutics, Inc., Nashville, TN, USA).

These three BoNT/A formulations are purified by protected proprietary procedures to homogeneity and, as claimed by the manufacturers, do not contain any bacterial proteins other than the 150 kDa BoNT/A. However, there are key differences in their excipients, which are added to the pure 150 kDa neurotoxin and present in the final product. These differences may affect the immunogenicity of the three CPF BoNT/A formulations.

### 3.2. General Considerations on Excipients in Pharma Protein Preparations

An excipient is defined as “an inactive substance that serves as the vehicle or medium for a drug or other active substance” (cited from [55]). “The reasons why excipients are added by the producers include long-term stabilization, bulking up solid formulations, to facilitate drug absorption, to reduce viscosity, or enhance solubility. In the manufacturing process excipients may confer non-stick properties, may aid stability in the vial by preventing denaturation or aggregation over the expected shelf life” (cited in modified form from [56]). Following this definition, excipients should not participate in the pharmacological mechanism for which the drug is designed. For BoNT/A formulations, this means that excipients should be inactive in neuromodulation. However, excipients may be active in another sense, as they may increase the immunogenicity of a pharma protein (reviewed in [57]).

One of the main reasons for adding protein-based excipients to BoNT/A formulations is to prevent the relatively low amount of BoNT/A protein (usually in the low nanogram (ng) range for a 100-unit vial) from forming aggregates or sticking to surfaces of the glass vials or syringes into which the reconstituted solution is drawn for injection. In addition, denaturation at the liquid/air interface is also prevented. To this end, human serum albumin (HSA) is frequently used as a “carrier protein” or “stabilizer”. Adding a relatively large amount of HSA (e.g., 1 mg per 100 U vial of INCO—see Table 1) to the stock solution of BoNT/A during formulation will ensure that HSA molecules are in large excess to BoNT/A molecules and will, thus, “coat” all surfaces and thereby prevent the active ingredient from being “lost”. In addition, HSA may interact with other proteins, as it does in blood, possibly ensuring an equal distribution of BoNT/A molecules within the solution. Finally, HSA has been discussed as possessing chaperon properties that help to maintain the correct fold of proteins in solutions ([58] reviewed in [59]). In summary, HSA functions as a multi-purpose stabilizer in pharma protein formulations.

If a manufacturer decides not to add HSA as a stabilizer, other proteins such as gelatine in one CPC BoNT/A formulation—Lantox [60]—or a peptide, as included in DAXI, may take over this function. It is also possible to formulate BoNT/A completely without peptides or proteins by adding detergents or surfactants, such as polysorbates, as in CORE and DAXI.

How dramatic the impact of an alteration of an excipient in a pharmaceutical can be has been known since the occurrence of red cell aplasia due to the induction of anti-drug antibodies (ADAs) in patients treated with a recombinant erythropoietin product (epoetin alfa = *Eprex^®^*), starting in 1998 until around 2003. One main difference of this immunogenic erythropoietin formulation was the substitution of human serum albumin with polysorbate 80 to avoid the hypothetical risk of virus or prion transmission by the human blood product. Although still not unambiguously resolved [61], it was proposed that in the presence of polysorbate 80, leached chemicals from the rubber stoppers used for the pre-filled syringes showed adjuvant properties, thus increasing the immunogenicity of the recombinant erythropoietin. This leaching did not occur when human serum albumin was present as a stabilizer [62,63,64]. Thus, closely examining the excipients in pharmaceuticals that contain the same active ingredient but differ in putatively “inactive” excipients seems justified, especially with respect to their immunogenic potential to induce the formation of anti-drug antibodies.

### 3.3. A Review of Excipients in CPF BoNT/A Formulations

The individual excipients for the three CPF BoNT/A formulations are summarized in Table 1 and will now be discussed in detail with respect to their potential direct or indirect influence on the immunogenicity of the 150 kDa BoNT/A.

#### 3.3.1. Human Serum Albumin (HSA)

Human serum albumin (HSA) has been in use as a plasma expander or part of pharmaceuticals for many decades (reviewed in [65,66]). Although HSA can be produced in recombinant form from non-human sources (reviewed in [67]), the vast majority of HSA included in pharmaceuticals is still of human origin, normally purified from human blood donations. Although concerns have been raised repeatedly that human blood products carry a very remote potential risk of being contaminated with human pathogens, such as small viruses (experience with human immunodeficiency virus in blood products) or prions (experience with BSE), HSA produced under well-controlled good manufacturing practice (GMP) conditions is considered to be safe (reviewed in [68]). Very few serious side effects have been reported despite the fact that HSA solutions are the most frequently used biopharmaceuticals, with literally hundreds of millions of doses applied since the mid-1940s (reviewed in [65,66]). HSA is also widely used as a carrier for certain drugs, e.g., in modern oncology (reviewed in [69,70]), due to its well-known characteristics and demonstrated low immunogenicity. It should be noted that the use of a human-derived protein in human beings has been discussed controversially for religious reasons [71] but is beyond the scope of this discussion.

From an immunological point of view, HSA in its native form (i.e., not denatured, unmodified) should be immunologically “inert”. HSA is present in very large amounts in human serum and even in tissue fluid. Normal concentrations of HSA in human blood or plasma are 35–50 mg/mL, and the concentration in the interstitial space (into which BoNT/A is normally injected) is reported to be around 20% to 60% of that in blood depending on the tissue type [72]. HSA as excipient does not differ in any way from the HSA of the person receiving injections; thus, the immune system of the person injected should be tolerant to HSA, although sporadic hypersensitivity reactions to HSA-containing pharmaceuticals have been reported previously [73,74,75]. The situation may be different if HSA is chemically modified, denatured, or certain haptens (antigens too small to cause an immune response themselves) bind to it. One prominent example is the contact of HSA-containing liquids with medical equipment that had been sterilized with ethylene oxide (reviewed in [76]). In such a situation, antibodies to this chemically altered HSA, now immunologically a neo-antigen, may arise. HSA is contained in many BoNT/A formulations on the market, including those longest in the market—ONA, ABO, and INCO—and its role has been discussed in great detail (reviewed in [66,77]). Reconstituting 100 units of INCO in the recommended 2.5 mL of saline results in 0.4 mg of HSA/mL, a concentration far less than is present in the tissue. Taken together, HSA can be considered a non-immunogenic excipient in biopharmaceuticals considering the vast number of applications worldwide and the extremely low number of reports on immune reactions. Furthermore, HSA, in its natural form, does not seem to contribute to the immunogenicity of a BoNT/A formulation.

#### 3.3.2. Sucrose

Sucrose is a non-reducing sugar, a disaccharide consisting of fructose and glucose that is contained in many foodstuffs. Sucrose is normally broken down into fructose and glucose in the gut and can then be metabolized in the body. Upon injection, it is normally excreted via the urine. Sucrose is an approved excipient in many pharmaceuticals [78,79]. It is used to stabilize pharma proteins in solution and during the processes of freeze drying. Although the lack of or excessive amounts of sugars have been described to affect the immune system in both stimulating or inhibiting ways, there are no reports that indicate sucrose, at the concentrations present in pharmaceuticals as an excipient, by itself causes immune reactions. It is also not known if it contributes to increasing immune reactions to other substances. In that sense, the excipient sucrose is immunologically inert. As a side note, it should be mentioned here that the so-called sucrose “allergies” are not real classical allergies. These adverse responses to sucrose intake are due to sucrose intolerance caused by the lack of the enzyme sucrase-isomaltase.

#### 3.3.3. L-Histidine/L-Histidine-HCl

L-Histidine is a naturally occurring essential amino acid, and L-Histidine Monohydrochloride is a salt of this amino acid. Histidine is a common excipient that is used to buffer solutions of biopharmaceuticals due to its pKa of 6 [80,81]. Recently, it was shown that Histidine may also prevent the aggregation of monoclonal antibodies in biopharmaceuticals [82]. The direct effects of Histidine on the immunogenicity of biologicals are unknown, and it can be considered immunologically inert.

#### 3.3.4. Trehalosedihydrate

Trehalose is a sugar, specifically a glucose disaccharide (Glc α(1→1)α Glc), that can be found in plants, insects, and fungi, where it serves naturally as an “anti-freeze“ agent (in cryptobiosis—the prevention of ice formation in living cells). As in nature, the addition of this sugar protects pharma proteins in freeze-drying processes. It prevents denaturation and protein aggregation during desiccation and also during renaturation [83,84]. However, trehalose, as part of glycans or glycolipids, has been shown to be immune-stimulatory, e.g., via Toll-like receptors [85], and capable of augmenting antibody formation [86]. One report showed that the addition of trehalose to a mumps vaccine augmented the immunogenicity of this vaccine in guinea pigs compared to the same vaccine containing HSA and/or gelatine as stabilizers [87]. Recently, other effects of trehalose on the immune system have been identified, such as the stimulation of autophagy [88]. However, presently, it remains unclear whether the trehalose concentration achieved after reconstitution of a vial of DAXI affects the immunogenicity of this CPF BoNT/A preparation.

#### 3.3.5. Polysorbate 20 (Polyoxyethylene (20) Sorbitan Monolaurate)

“Polysorbate 20 … is a polysorbate-type nonionic surfactant formed by the ethoxylation of sorbitan monolaurate. Its stability and relative nontoxicity allows it to be used as a detergent and emulsifier in a number of domestic, scientific, and pharmacological applications. … the ethoxylation process leaves the molecule with 20 repeat units of polyethylene glycol; in practice these are distributed across 4 different chains, leading to a commercial product containing a range of chemical species.” (citation modified from [89]). Polysorbate 20 is used as a surfactant or nonionic detergent (also known as Tween 20) in foodstuffs to increase solubility. In scientific applications (biochemistry), it is employed to dissolve cells and solubilize membrane proteins. As a surfactant, it binds to proteins and helps to prevent the aggregation of proteins and their adsorption at interfaces (hence being used in buffers in ELISAs, Western blots, etc.) [90]. Although the exact mechanisms by which polysorbates stabilize proteins in solution are not known, interfacial competition and surfactant–protein complexation have been discussed as the two main mechanisms [91]. “Addition of polysorbates prevents protein unfolding at the interface during the manufacturing process, sample handling and storage, including mixing, filtration, pumping, shaking, agitation, and freeze-thaw. Similarly, it can also prevent protein adsorption and subsequent loss at the product contacting surfaces, such as filters, primary container/closures, and IV administration tubing, playing a critical role to assure accurate dose delivery to patients” (citation modified from [91,92]).

Two aspects of polysorbates have to be discussed with respect to their possible effects on the immunogenicity of biologicals: (1) their direct adjuvant properties and (2) their possible indirect effect on proteins due to their propensity to auto-oxidize and form radicals. “Since the polysorbate degradation may inadvertently affect the quality, efficacy, safety, and stability of the protein formulation, there is increasing scrutiny from health authorities on polysorbate control strategies to assure that polysorbate content remains constant during shelf life of drug products” (citation from [91]).

First, polysorbates have been included in adjuvants for different vaccines (reviewed in [93]). They are contained in several approved emulsion-based adjuvants, one prominent example being MF59, a mixture of 5% squalene, 0.5% polysorbate 80, and 0.5% sorbitan trioleate (reviewed in [94]). The exact mechanisms by which these adjuvants increase the immune response to vaccines are not completely understood. However, it seems that emulsion-based adjuvants result in a very strong and early activation of the innate immune system. In addition, stronger activation of naïve B lymphocytes and longer-lasting B lymphocyte responses have been observed (reviewed in [95]). In general, polysorbates in emulsion-based adjuvants are pro-inflammatory, as they participate in the initiation of an acute inflammatory response at the site of injection, resulting in antigen uptake and presentation, and the promotion of an adaptive immune response, including antibody production. Whether polysorbates alone can activate immune cells is unclear, although hypersensitivity reactions have been reported (reviewed in [96]). Second, polysorbates may undergo auto-oxidation and subsequent degradation, giving rise to radicals, including reactive oxygen species ([97] reviewed in [98,99,100]). Such reactive products can chemically modify proteins [91], which leads to new epitopes (neo-antigens) that could increase the immunogenicity of the pharma protein. A further issue is that commercial polysorbate 20 batches may differ in their quality, as polysorbate 20 is known to consist of a mixture of subspecies and even byproducts. They may also differ in one major characteristic—their critical micelle concentration (CMC)—with consequences for their stabilizing effects against aggregation and particle formation in biopharmaceuticals [101].

Little has been published on polysorbates in BoNT/A formulations. Recently, the safety of polysorbate 20 was tested in rats and rabbits [102]. Under these conditions, the authors claim that polysorbate 20 demonstrated a low immunogenic safety profile comparable to HSA.

As both histidine and polysorbate 20 are excipients in DAXI, it is of interest to note that histidine may affect polysorbate 20 degradation in two (opposing) ways. First, L-Histidine may scavenge reactive oxygen species and, thus, protect the pharma protein in solution. On the other hand, it may also accelerate polysorbate 20 oxidation during storage in solution [103,104], e.g., if not all reconstituted material is used immediately as recommended by the manufacturers. Whether or not this plays a role in the CPF BoNT/A formulation is presently unclear.

#### 3.3.6. RTP004 Peptide

The RTP004 peptide in DAXI is a synthetic 35-amino-acid peptide with a calculated molecular mass of 4.826 kDa [105]. A part of the sequence (RKKRRQRRR) is derived from the human immunodeficiency virus (HIV-1) TAT protein in which this sequence allows the HIV protein to penetrate the plasma membranes of target host cells ([106] reviewed in [107]). Therefore, it is also called a cell-penetrating peptide (CPP) or protein transduction domain (PTD).

In PCT patent application WO 2008/082885 “*Transport molecules using reverse sequence hiv-tat polypeptides*”, filed in 2007, Revance Therapeutics Inc describes, among others, the use of this CPP to transport botulinum neurotoxin through the skin ([108] (reviewed in [109]). The status of this patent as of June 2023 is “abandoned”. However, to date, there is no approved or commercially available botulinum toxin formulation that utilizes the principle of a cell-penetrating peptide to allow the transport of the relatively large cargo molecule, i.e., BoNT/A (150 kDa), through the skin.

In DAXI, a BoNT/A formulation that is applied “traditionally” via injection, this HIV TAT-derived sequence appears twice in RTP004, at the N- and C-termini, with two adjacent glycines to a central linker of lysines as a poly-lysine core (RKKRRQRRRG-K_15_-GRKKRRKKQRRR). In total, the resulting peptide is highly positively charged at physiological pH (31 amino acids positively charged out of 35 amino acids in total). The manufacturing company states that it is a “novel excipient” that forms strong electrostatic bonds with BoNT/A. This proprietary stabilizing peptide is said to allow the formulation of DAXI without human serum albumin and is reported to stabilize the product at room temperature. Thus, the cell-penetrating property of RTP004 does not seem to play a role in the molecular mechanism of neuromodulation, although it is still designated a “*unique proprietary protein transduction domain (PTD) excipient*” (cited from [110]). The manufacturer claims that the highly positively charged peptide binds non-covalently, but tightly, to the negatively charged BoNT/A proper and thereby stabilizes the pharma protein to prevent protein aggregation and adsorption to surfaces [111]. It has been suggested that the strong positive charge of the RTP004 peptide promotes binding to negatively charged neuronal surfaces and extracellular matrix proteins, possibly increasing the internalization of the neurotoxin by an unknown mechanism [110]. If that is true, RTP004 has to be considered both an excipient and an active ingredient [110,112] as stated by the manufacturer [11]. However, it remains unclear why the binding of the positively charged RTP004 peptide should be selective for negatively charged neuronal surfaces, as practically all cell types of the body are negatively charged due to the terminal sialic acid residues on cell surface glycoproteins [113].

#### Possible Effects on Immunogenicity

In DAXI, RTP004 is added to BoNT/A in a large molar excess. Thus, it can be postulated that several RTP004 peptides will bind to negatively charged areas on the surface of BoNT/A, creating novel surface structures on BoNT/A heavy and/or light chains. It can be hypothesized that these novel charged surface areas might be recognized by the immune system as neo-epitopes, in addition to the epitopes of the antigens BoNT/A and RTP004 peptide, respectively. Even if RTP004, due to its size of less than 5 kDa, would behave as a hapten, it may elicit an immune response as it sticks to BoNT/A, which could behave as a carrier enabling an immune response also to RTP004. Thus, the issue of immunogenicity of RTP004 alone and BoNT/A alone, as well as of the non-covalently linked complex of BoNT/A plus RTP004, becomes relevant with respect to this CPF BoNT/A product. This issue was partially addressed in a first “multi-study” evaluation of pooled data from the SAKURA Phase 3 clinical trials with DAXI [110,114]. No neutralizing antibodies to BoNT/A were found in a total of 2786 subjects, while treatment-related anti-BoNT/A antibodies were found in 0.8% of patients and anti-RTP004 antibodies were detected in 1.3% of patients without impact on treatment (glabellar lines at four weeks after treatment cycle). In any case, it was demonstrated that RTP004 is an immunogenic peptide. It is not clear whether the peptide is immunogenic, per se, or a hapten by binding to carrier proteins (BoNT/A or other endogenous proteins to which RTP004 binds after injection). The authors claim that the low antibody rates had a negligible effect on DAXI efficacy or immune-related effects [110,114]. However, one *caveat* of this evaluation is that of the 2786 subjects, 882 received two treatments and only 568 received three treatments; thus, the overall timeframe for the development of antibody-mediated SNR may have been too short to draw robust conclusions.

#### 3.3.7. L-Methionine

L-Methionine is a sulfur-containing naturally occurring essential amino acid. It is an antioxidant that can be used to protect methionine and tryptophane residues in pharma proteins from oxidation [115,116]. The oxidation of methionine or tryptophane residues in proteins may increase their immunogenicity as this creates neo-epitopes (reviewed in [117]), possibly also in BoNT/A. Methionine is normally added in large excess to a solution of pharma proteins. In the presence of oxidizing agents during processing or storage, it serves as a surrogate substrate and becomes oxidized to methionine sulfoxide. In addition, it also participates in buffering a solution.

#### 3.3.8. NaCl

Sodium chloride is a salt (“cooking salt”) and, by itself, is, of course, not immunogenic. However, Na^+^ and, especially, Cl^–^ ions may affect the conformation of proteins depending on their concentration in solution [118]. Although NaCl is generally considered a non-denaturing salt, high salt concentrations tend to destabilize non-covalent interactions between amino acid residues and affect intra- and inter-molecular bonds in proteins. These include electrostatic interactions, hydrogen bonding, hydrophobic interactions, and van der Waals interactions. Salt bridges within proteins may also be affected [119], especially if they are exposed on the surface of the protein [120]. This may have a chaotropic effect, resulting in the destabilization and, possibly, denaturation of proteins, especially when a biopharmaceutical is vacuum-dried or lyophilized for stable storage (reviewed in [121]). This comes into effect when water is removed from a protein solution for better stability. Salt concentrations will increase with the removal of water. After reconstitution of the vacuum- or freeze-dried pharma protein, some of the wrongly folded proteins may not fold back to their proper native conformation, resulting in inactive proteins. These inactive proteins tend to form aggregates due to exposure to hydrophobic amino acid residues or hydrophobic pockets within the protein to water. Protein aggregates may be more immunogenic than proteins in their native conformation (see below), causing antibody formation against the protein, of which some may be neutralizing. The addition of ionic salts, such as NaCl, is considered critical in a lyophilization process [122] because it may also negatively affect the freeze-drying process. In some cases, however, the addition of NaCl improves the appearance of the cake of the freeze-dried materials without affecting its bioactivity [123]. For BoNT/A, it was reported that the presence of NaCl during the freeze-drying process results in a loss of bioactivity [124].

## 4. Potential Influence of Excipients on the Specific Bioactivity of BoNT/A Products and the Relevance for Immunogenicity

### 4.1. Specific Bioactivity of BoNT/A Formulations

Specific bioactivity describes the bioactivity of a protein product related to the mass of the protein (e.g., units per ng protein). The specific bioactivity is determined by the ratio of the number of BoNT/A molecules per mass (or in a vial) that work as neurotoxins—and are, thus, bioactive—to the number of BoNT/A molecules per mass that are not working as neurotoxins because they have been denatured during the purification process, lyophilization, storage, or reconstitution and, thus, are inactive. The specific bioactivity is highest when no inactive molecules are present in the formulation. As the specific bioactivity of a BoNT/A formulation is relevant for its immunogenicity (see below), it is of great importance to address the question if all three CPF BoNT/A formulations contain the same number/concentration of bioactive BoNT/A molecules per mg/or vial.

Relevant for neuromodulation are only those BoNT/A molecules that are bioactive. This means that they must be able to bind to their specific receptors on the surface of the nerve terminal to facilitate uptake into the recycling vesicles. An intact heavy chain of the 150 kDa BoNT/A is responsible for this process. The disulfide bridge between heavy and light chains must be intact; otherwise, the light chain will not be taken up into the recycling vesicle and will not be translocated into the cytosol of the nerve cell. The light chain then has to reach the cytosol of the nerve terminal to finally cleave SNAP25. Only this proteolytic activity of the light chain abrogates the fusion of the neurotransmitter-filled vesicle with the plasma membrane of the nerve cell at the nerve terminal (reviewed in [125]). In summary, heavy and light chains must be correctly folded in their three-dimensional structure, and one light chain must be covalently linked to one heavy chain by a disulfide bridge to be bioactive as a neurotoxin.

### 4.2. Measuring Protein and the Relative Content of Active and Inactive BoNT/A Molecules

To measure the correct three-dimensional structure or folding of a protein is not trivial. It can only be achieved either by sophisticated biochemical/biophysical techniques usually employing large amounts of BoNT/A or, indirectly, by measuring the neuromodulatory effect in a cell-culture- or animal-based bioassay, e.g., the “classical” mouse LD-50 assay. This determines “biological units” per amount of BoNT/A protein or the specific bioactivity of a preparation in units/mg BoNT/A protein. As different companies use different bioassays to determine “the bioactivity of their product”, individual units cannot be directly compared and do not necessarily mean the same. For a fair comparison, the different preparations need to be analyzed in a head-to-head assay. It would be necessary and helpful to standardize these “company” units to a common BoNT/A standard as has been conducted in the past, e.g., for recombinant cytokines [126].

In addition, the exact protein concentration of the sample has to be known to define the specific bioactivity. Due to the extremely high potency of BoNT/A, this can only be performed reliably during the production process when highly concentrated stock solutions are available. It is practically impossible to conduct this with the minute amounts of the final product contained in a commercial vial. In addition, the presence of other proteins or peptides as stabilizing excipients will interfere with protein measurements. This is especially true for INCO in which human serum albumin is contained and DAXI in which the RTP004 peptide is added at high concentrations. An alternative for determining the concentration of BoNT/A in a vial, in the presence of excipient proteins or peptides, is to employ an enzyme-linked immune-sorbent assay (ELISA) specific for BoNT/A. This is a highly sensitive immunological technique based on antibodies, usually antibody pairs, specific for BoNT/A. However, it must be stressed that ELISAs are normally unable to distinguish between active and inactive proteins. They only recognize the amount of antigen, frequently irrelevant to the correct three-dimensional structure. In addition, especially if monoclonal antibodies are used, it is relevant to know the exact antigenic epitope that is recognized by the antibodies used in the ELISA because they might bind only either to the heavy or the light chain, irrespective of their linkage by the disulfide bridge. The analysis can also be affected if an epitope is modified in a product. Thus, an ELISA might also yield misleading results if there are structural differences.

Presently, only the proportion of bioactive BoNT/A molecules relative to inactive BoNT/A molecules for INCO has been published in the scientific literature, with very high specific bioactivity (0.44 ng/100 U = 4.4 pg/U) reported [127]. The specific bioactivities of DAXI and CORE have not been disclosed. This discussion may seem academic and irrelevant to physicians and patients as long as the manufacturers fill enough bioactive BoNT/A molecules into the vial to achieve the activity in units claimed on the vial (and paid for). This is true for achieving the desired clinical effect. Yet, the presence of inactive BoNT/A molecules may be critical with respect to the immunogenicity of the formulation.

### 4.3. Inactive Proteins Are Not Irrelevant in a BoNT/A Formulation as They May Increase Its Immunogenicity

In the field of biopharmaceuticals, inactive proteins are known to cause problems by increasing the immunogenicity of the product and, thus, increasing the risk of antibody formation to the drug, i.e., anti-drug antibodies (ADAs). Unfortunately, these may include neutralizing ADAs that interfere with therapy or even abrogate the desired effect. One reason for this is that inactive proteins are wrongly folded and/or may even be proteolytically cleaved. This frequently results in the aggregation of misfolded, denatured, or cleaved proteins due to the thermodynamic instability of exposed hydrophobic regions in aqueous solutions (water). Protein aggregates have long been recognized as being one main reason for the increased immunogenicity of biologicals [128,129,130,131,132]. Protein aggregates normally do not occur in the interstitial fluid (the extracellular space) in our body and, thus, are recognized by DCs as “endogenous danger signals” [133,134]. Therefore, they may behave like adjuvants by providing the “danger signals” as discussed above. In consequence, DCs become activated and initiate an antigen-specific immune response that may finally result in the formation of antibodies, including neutralizing antibodies against BoNT/A.

Neither the amount of inactive BoNT/A molecules nor the concentrations of possible protein aggregates in reconstituted solutions of the three CF BoNT/A formulations are known or published. However, based on decades of experience with other pharma proteins, it can be assumed that inactive or denatured BoNT/A molecules and their breakdown products, both capable of forming protein aggregates, may exist in formulations that do not have very high specific bioactivity. Critical steps that may affect the bioactivity of proteins include the entire purification process and, most prominently, the removal of water for stable storage, e.g., by lyophilization and the reconstitution of desiccated pharma proteins prior to clinical use. Both steps may result in the denaturation of a proportion of the proteins, even if the initial stock solution does not contain detectable amounts of inactive or denatured proteins or if measures have been taken to avoid denaturation by the addition of stabilizers. It needs to be emphasized that if the concentration of protein aggregates in a solution is low, these aggregates may not be ascertained by conventional analytical methods such as turbidimetry or analytical ultracentrifugation. Yet, they may still be “sensed” by dendritic cells, probably the most sensitive cell type for detecting “irregularities” in our body, including protein aggregates [135].

Thus, indirectly, the issue of the specific bioactivity of a BoNT/A formulation (as high as achievable and containing as few inactive neurotoxin molecules as possible) moves out of the realm of the manufacturing pharmaceutical companies and becomes relevant to physicians and patients in daily practice. It also warrants attention in order to reduce the risk of generating ADAs.

## 5. Conclusions

Presently, three BoNT/A products are free of complexing proteins and contain only the 150 kDa neurotoxin molecule. Yet, they are not the same. They differ in proprietary production protocols and their excipients, which may affect the individual immunogenicity of the products. INCO contains only two immunologically inconspicuous excipients and has demonstrated low immunogenicity in daily practice and many well-controlled clinical studies over more than ten years. DAXI contains polysorbate 20 and the highly positively charged RTP004 peptide, which have the potential to increase the immunogenicity of BoNT/A by introducing neo-epitopes. Results from early clinical studies show antibodies against BoNT/A and RTP004 at low frequencies. However, the follow-up period is critically short with a maximum of only three injections. CORE contains polysorbate 20 and NaCl, which may also influence the specific bioactivity and immunogenicity of the product. Presently, no data are published on CORE’s immunogenicity in patients.

Robust clinical studies with long-term follow-up periods are required to ascertain the effect of these excipients on the immunogenicity of each CPF BoNT/A product. The future will tell whether or not all three CF-BoNT/A products demonstrate the same low immunogenicity in daily practice in a large number of patients, despite the differences in excipients and production protocols.

## Figures and Tables

**Figure 1 toxins-16-00101-f001:**
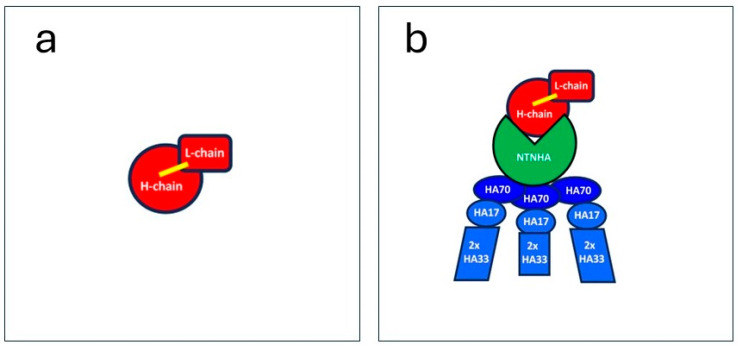
Schematical depiction of botulinum neurotoxin A. (**a**) The 150 kDa neuromodulator BoNT/A consists of one heavy chain of 100 kDa (red circle) and one light chain of 50 kDa (red square) covalently linked by a disulfide bridge (yellow line). (**b**) The progenitor complex of 900 kDa is composed of the 150 kDa neuromodulator BoNT/A (red), a non-toxic non-hemagglutinin (NTNHA) of 150 kDA (green), and a 12-subunit complex consisting of 3 different hemagglutinins (HAs): 3 times HA70, 3 times HA 17, and 6 times HA33. The schemes were modified from the structures in [3].

**Figure 2 toxins-16-00101-f002:**
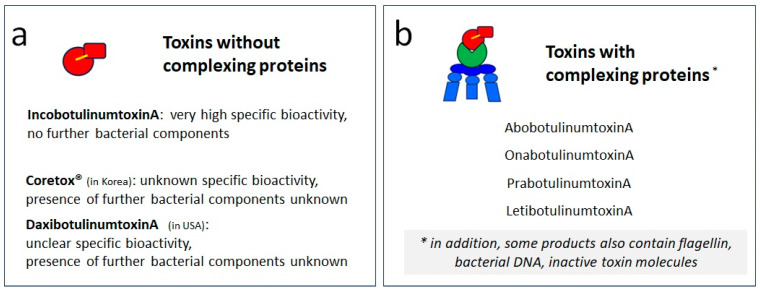
Schematic depiction of principal differences in purity of selected BoNT/A preparations. (**a**) BoNT/A formulations without bacterial complexing proteins, containing only the 150 kDa neuromodulator, namely IncobotulinumtoxinA [8,9], Coretox^®^ [10] and DaxibotulinumtoxinA [11,12]. (**b**) BoNT/A formulations containing clostridial complexing proteins, such as AbobotulinumtoxinA [13], OnabotulinumtoxinA [14], PrabotulinumtoxinA [15] and LetibotulinumtoxinA [16]. Some of these also contain bacterial components such as flagellin, bacterial DNA and inactive toxin molecules [17,18].

**Figure 3 toxins-16-00101-f003:**
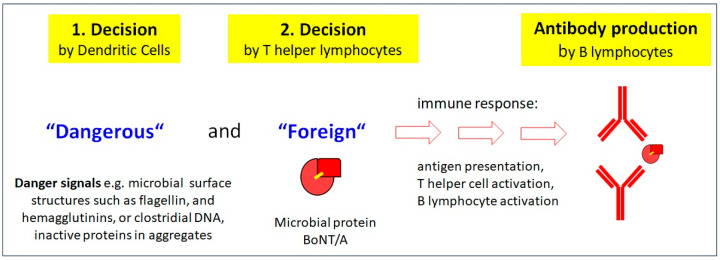
Simplified model of the activation of the immune system. Two decisions have to be made by the immune system in a strictly hierarchical order before antibodies can be produced. Firstly, sentinel dendritic cells need to be fully activated by “**danger signals**” to become professional antigen-presenting cells. They present peptides of the antigen (here, BoNT/A) to an antigen-specific T helper lymphocyte, which can recognize the presented peptide as “**foreign**”. Secondly, if the peptide is foreign, such as in clostridial BoNT/A, this T helper lymphocyte becomes activated and subsequently supports antigen-specific B lymphocytes. These become activated and finally produce and release antigen-specific antibodies to BoNT/A, possibly including neutralizing antibodies.

**Table 1 toxins-16-00101-t001:** Differences in excipients in the three BoNT/A products free of bacterial complexing proteins.

Product Name	IncobotulinumtoxinA	MT-10107	DaxibotulinumtoxinA-Lanm
Trade name	Xeomin^®^	Coretox^®^	Daxxify^®^
Manufacturer	Merz, Frankfurt, Germany	Medytox, Seoul, Republic of Korea	Revance, Nashville, USA
Dosage	50 U, 100 U	100 U	50 U, 100 U
Toxin type	*C.bot.* type A (Hall)	*C.bot.* type A (Hall)	*C.bot.* type A (Hall)
Active ingredient (s)	Core 150 kDa toxin	Core 150 kDa toxin	Core 150 kDa toxin RTP004 peptide (11.7 µg)
Appearance	Lyophilizate	Lyophilizate	Lyophilizate
Storage	Room temperature	2 °C to 8 °C, refrigerate	Room temperature
Formulation (excipients)	Human serum albumin (1 mg) Sucrose (4.7 mg)	L-Methionine (?) Polysorbate 20 (?) Sucrose (3 mg) NaCl (0.9 mg)	L-Histidine (0.14 mg) L-Histidine-HCl monohydrate (0.65 mg) Polysorbate 20 (0.1 mg) Trehalosedihydrate (36 mg)
“Stabilizer”	Human serum albumin	Polysorbate 20	RTP004 peptide Polysorbate 20
References	[8,9]	[10]	[11,12]

## Data Availability

Not applicable.

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
