# Peer review of "Complexing Protein-Free Botulinum Neurotoxin A Formulations: Implications of Excipients for Immunogenicity"

_toxins, 2024, doi:10.3390/toxins16020101_

Round 1

Reviewer 1 Report

Comments and Suggestions for Authors

This work compares the protein-free botulinum neurotoxin A (BoNTA) formulations with regards to their respective excipients. The author also provides adequate background information on complex-free BoNT formulations and the immunogenicity related to protein versus protein-free formulations. Overall, this work is a good review of BoNTA excipients.

Comments on the Quality of English Language

Minor grammatical edits are required.

Reviewer 2 Report

Comments and Suggestions for Authors

The review has been performed well. The manuscript is extensive and interesting. The overall composition of the manuscript is good. I recommend acceptance of the paper with minor points.

1) The authors need to reinforce their focus on why they carry out this work. The way as presented is weak.

 2) Abstract section. DAXI and CORRE mention that contains four excipients; however, only two are described. Please introduce the other two excipients.

 3) In Figure 1, please change the order of panel B because in the main document Figure 1B is first described and then panel A. Additionally, on line 43 Figure 1 must be cited in the main document.

 4) In all figure: Merge the Figure title with Figure legends. Besides, all figures must be improved. They are seen in low resolution and the way they are presented is very simple.

 5) The table must be improved. In the current format, it does not have the appropriate resolution. Additionally, a table and not an image must be provided. Please verify. Human eerum? or Human serum?

 6) Lines 152 and 154: please change nAb by nAbs. Please verify all the main document.

 7) 2.3. Why and how the immune system responds to injections with BoNT/A: Lines 154-190 are repetitive with lines 195-220. Please remove the second paragraph.

 8) Line 234. Remove the point in the title. Verify in the main document.

 9) 3.1. Free of complexing proteins – yet not the same: Please modify the title, the way it is presented is difficult to understand.

 10) Lines 236-240 and lines 246-256 were previously described. Please check and remove it, this is repetitive.

 11) Finally, the manuscript contains many typographical and grammatical errors. I suggest authors carefully review the entire main document.

Reviewer 3 Report

Comments and Suggestions for Authors

Dear Authors,

Title is too long maybe beneficial to reduce it

 Abstract :

 Line 7 : as it may result in "secondary treatment failure"

 Introduction :

 Line 42 : please also mention the presence of two components of the heavy chain : HC and HN, one for the binding and one for the translocation

 Line 45 : please correct : the toxin can cross intestinal barrier and cause paralysis without injection by entering the blood stream

 Line 55 : hemaglutinin

 Line 83 : bacterial culture (not cells)

 In figure a, please explain how bacterial contaminants can be present and what they may be ?

 Line 8 of the table : serum albumin.

 Line 115 : production protocol is standardised and not "insufficient", it is just difficult to modify the production protocol without altering the active product BoNT/A.

 Section 2.2 , Line 38 : the tetanus vaccines contain specific adjuvants helping develoment of the immunity, this is not the case in BoNT/A products so this statement is controversial. Please amend the sentence.

Line 164 : remove or replace « really »

 Please also introduce briefly the rôle of macrophages

 Line 210 to 220 : Please correct this statement since the BoNT/A itself can induce immunisation as the heavy chain is used for vaccination and antibody production. Also the LC can be used.

252 : It is not possible to assume that the Hall strains are the same since there are no available proof, so the differences may also come from the BoNT/A nature itself.

 Line 342 : This statement should be modulated since HSA can be contaminated and are immunogenic even if this is rare.

 BoNT/A toxins standards are available commercially with precise specific activity measurements.

Section 4.2. Please also mention the existence of the standard mouse bioassay that is used to calibrate the BoNT/A in LD50 units globally and since decades. Also, please mention the cell based assays developments.

Comments on the Quality of English Language

The manuscript is well written, only minor editing would be needed to clarify a few long sentences.
